# Stem Cells in Cardiovascular Diseases: 30,000-Foot View

**DOI:** 10.3390/cells10030600

**Published:** 2021-03-09

**Authors:** Thomas J. Povsic, Bernard J. Gersh

**Affiliations:** 1Department of Medicine, and Duke Clinical Research Institute, Duke University, Durham, NC 27705, USA; 2Department of Cardiovascular Medicine, Mayo Clinic College of Medicine, Rochester, MN 55905, USA; Gersh.Bernard@mayo.edu

**Keywords:** stem cells, regenerative medicine, clinical trials, congestive heart failure, cardiomyopathy, angina

## Abstract

Stem cell and regenerative approaches that might rejuvenate the heart have immense intuitive appeal for the public and scientific communities. Hopes were fueled by initial findings from preclinical models that suggested that easily obtained bone marrow cells might have significant reparative capabilities; however, after initial encouraging pre-clinical and early clinical findings, the realities of clinical development have placed a damper on the field. Clinical trials were often designed to detect exceptionally large treatment effects with modest patient numbers with subsequent disappointing results. First generation approaches were likely overly simplistic and relied on a relatively primitive understanding of regenerative mechanisms and capabilities. Nonetheless, the field continues to move forward and novel cell derivatives, platforms, and cell/device combinations, coupled with a better understanding of the mechanisms that lead to regenerative capabilities in more primitive models and modifications in clinical trial design suggest a brighter future.

## 1. Introduction

The heart is traditionally considered a post-mitotic organ with limited, if any, reparative capacity. Thus, the idea that cellular-based approaches might renew, rejuvenate, and regenerate the heart is immensely appealing. The allure of such approaches is augmented by observations that heart regeneration not only is possible but occurs in more primitive organisms [1,2] and in fetal mammalian development [3]. While considerable effort is being expended to better understand the mechanism(s) whereby organisms such as zebrafish effect cardiac regeneration, initial clinical development was driven by the rather surprising discovery that many adult tissues, including the post-mitotic organs like the heart, house resident stem cells that could transdifferentiate into mature cells [4,5,6]. This led directly to the study of the use of readily available stem cell sources to enact cardiac repair with some dramatic and perhaps surprising reports of improvements in left ventricular function [7]. These experiments, which in some cases appeared inconsistent with clinical experience, raised several questions including the question of why bone marrow cells, which circulate through end-organs naturally at many times the doses administered in these studies, would affect such effective regenerative improvements. Several groups were subsequently unable to reproduce these findings [8,9]. In addition, while transdifferentiation of bone marrow precursors to endothelium and the potential for neovascularization appears reproducible, the capacity for regeneration of new cardiomyocytes from these sources appears extremely limited and may be due to fusion with native cells [8,9,10,11]. A second point of controversy was the degree to which cardiac stem cells might be capable of actively regenerating the adult heart. While several lines of evidence point to a low-level ability of the heart to generate new myocytes, significant myocardial regeneration is belied by clinical experience in which clinically significant “regeneration” is not observed after myocardial infarction, especially in the elderly and chronically diseased. Given this discrepancy between some early pre-clinical studies and the experience in clinical practice, it is perhaps not surprising that the field of cardiovascular cell therapy has largely been perceived as a disappointment, with an inability to translate to clinically effective approaches and fulfill the considerable hope (or hype) raised by early pre-clinical and clinical experiments [12].

This review seeks to understand and contextualize the field. We aim to provide a 30,000-foot overview of our perception of how the field has developed, appraise choices (and perhaps mistakes) made to date, and finally to discuss paths forward and promising technologies that may yet fulfill the promise of regenerative approaches to treatment of heart disease. 

## 2. Background

The lure of a “fountain of youth,” a mythical water capable of restoring the vitality of youth, dates back many centuries and is imbedded in the cultures of civilizations throughout the world, and is especially prominent in cultures during periods of exploration and expansion. It is therefore not surprising that the idea of using cells capable of tissue regeneration captures the imagination and fancy of researchers as well as patients and the lay public. In this century, the development of bone marrow transplantation has realized its potential as a curative therapy for various hematopoietic diseases. Given these possibilities, might not stem cells also be used to treat other organs, including the heart and brain, two organs traditionally considered “post-mitotic” and incapable of repair? 

At least three lines of evidence greatly increased the hope of stem cell-mediated cardiac repair (Figure 1). 

The first was the demonstration that primitive organisms, such as zebrafish [1,2,15] and mammals in fetal or neonatal stages of development [3], possess marked cardiovascular regenerative capacity far beyond what is observed in humans. In these models, almost complete recovery after even a massive amount of myocardial injury (~25% of left ventricular mass) with histologically indistinguishable tissue was observed [2]. Whether or not the signaling pathways (software) and cellular components (hardware) that allow this type of regenerative capacity can be adequately characterized, reactivated and to what extent they are even present in humans remains to be determined. 

The second intriguing line of early investigation was a series of experiments that suggested that cardiac regeneration occurs in the adult heart. Several lines of evidence pointed to this possibility. In human cardiac transplant recipients, careful (albeit technically difficult) experimentation established that myocytes of host origin might be found in the transplanted organ [14,16,17,18]. The converse experiment, in which patients received gender mismatched bone marrow transplants, confirmed the presence of rare cardiomyocytes of host bone marrow origin [14,19]. These experiments are technologically challenging and it is difficult to absolutely exclude other explanations, such as the possibility that cells identified as host cells are endothelial or vascular cells that are actively dividing and continually being replaced, which some studies have suggested [20]. The technical challenges associated with these experiments may at least in part account for the widely disparate degree of myocardial replacement reported in these studies (from <0.02% to upwards of 9%) [16,18], although other factors (including intentional falsification) may have played a role. Notably the majority of studies suggested that (1) cardiomyocytes of host origin were rare (<0.16%) [16,17,19] and (2) that non-myocytes such as endothelial and smooth muscle cells of host origin were significantly more common, suggesting that vascular turnover occurs more frequently [16,17,18].

A completely different line of investigation also supported the active replacement and regeneration of adult myocardium by host tissues. Taking advantage of the variation in atmospheric ^12^C levels in the 20th century as a result of atomic testing performed mid-century, Bergman and colleagues elegantly demonstrated low level cardiomyocyte replacement that declines with age [13]. While these researchers faced technical challenges similar to those in the transplant experiment (namely isolating cardiomyocytes without cross-contamination), their findings were consistent with previous results. This consistency lends credence to their research and suggests that cardiomyocyte regeneration likely occurs, although not at clinically meaningful levels. However, their results offer hope that the mechanisms could be identified and the regenerative response could be augmented. 

Finally, pre-clinical (and some early clinical) studies raised hopes of a speedy pathway to early success. Some highly cited publications reported significant myocardial regeneration and recovery in left ventricular function in animal models of acute infarction after treatment with isogenic bone marrow cells [4,5]. Other experiments used bone marrow cells to affect regression of atherosclerosis [21], with potentially transformative implications [22]. Unfortunately, follow-up experiments frequently failed to replicate these strongly positive results [8,9].

## 3. Early Development in Acute Myocardial Infarction

A direct consequence of these studies was an urgency to investigate translation to the clinical arena. Given their ease of acquisition, autologous nature, and with the backing of pre-clinical studies, unselected bone marrow cells received the preponderance of early attention. While some early studies suggested positive results in acute myocardial infarction [23], a series of trials drew more tempered conclusions based on treatment effects that were moderate at best [24,25]. These phase II trials suggested some evidence of efficacy, but the results were not uniformly reproduced; however, in aggregate, there were suggestions of modest benefits on both imaging and clinical endpoints [26,27,28,29,30,31]. While a phase III mortality trial was undertaken after immense pan-European efforts [32], the difficulty in obtaining funding, overcoming regulatory hurdles, and the overall cost of the program led to enrollment of only a small fraction of that originally planned, making the final results inconclusive [33]. In addition to these logistical factors, the results of the Bone Marrow for Acute Myocardial Infarction (BAMI) trial reflected the risk and uncertainty inherent in clinical research. While stem cell therapy might be a conceptually appealing approach to prevention of remodeling in this acute setting, the massive shifts in processes of care for ST segment elevation myocardial infarction have led to improvements in outcomes that made the original statistical suppositions of the trial outdated [34], making demonstration of any clinically meaningful benefit unlikely and impractical. 

Several lessons might be learned from these experiences. First, some of the early pre-clinical experiments that formed the foundation for this line of research were later found to be difficult to replicate or over-interpreted [8,9]. Research that appears to be “too good to be true” or inconsistent with clinical experience needs to be, and has been, questioned. Second, basing clinical programs on expectations of overly optimistic treatment effects generally leads to disappointment [35,36,37], particularly when this takes place on a backdrop of a changing and improving natural history. In hindsight, it was unlikely that an unselected bone marrow product in which stem cells are only a minor component would produce large improvements in outcomes. Finally, even the best of ideas may succumb to unanticipated changes and advances in both science and clinical medicine. With regard to acute myocardial infarction, advances in the delivery and nature of reperfusion therapy accompanied by gratifying and substantial improvements in outcomes has led to immense difficulty in demonstrating clinical improvements in patients who are now treated with markedly improved systems of care and expeditious primary percutaneous coronary intervention [38,39]. Even amongst patients who present with shock and survive to a time point at which stem cells are conventionally administered, the risk of myocardial infarction, cardiac death, or re-admission for heart failure is relatively low [40]; thus, it no longer appears that acute myocardial infarction is an attractive target for regenerative approaches. 

Despite these lessons, these pioneers accomplished much to establish, nurture, and invigorate a line of investigation that laid the foundation for the future of cardiovascular regeneration. While this approach has not brought immediately useful clinical therapies, we should understand that first generation products rarely represent the ultimate most effective approach, and that most breakthroughs reflect a series of improvements based upon the building blocks and lessons of prior research. 

## 4. Next Generation Products: The Middle Years

The last decade has witnessed a host of studies investigating various cellular approaches to treatment of a broader range of cardiovascular diseases. Rather than attempt to summarize or tabulate individual studies, we seek in this high-level overview to encompass key themes that we see driving this field of research over the last decade(s). 

### 4.1. Transition to Allogeneic Products

Autologous products were initially utilized based on ease of procurement and local manufacture, lack of concern for issues such as rejection and immune-mediated inflammatory responses, and a theoretically better safety profile, given that treatment consisted of cells that are already present in our bodies and/or circulate freely. Nonetheless, there are several advantages of an allogeneic product that have fueled a gradual transition to greater investigation of these cell sources. 

Allogeneic cells can be grown in large quantities and formulated as an “off-the-shelf” product, avoiding an invasive, sometimes surgical, harvesting procedure. An allogeneic product can be better characterized as a master cell line which can be tested in a variety of ways for potency, purity, and safety issues (contamination, genetic stability). Allogeneic cells obtained from “healthy,” often younger, donors may exhibit better functional capacity than autologous sources acquired from elderly patients with chronic vascular disease [21,41,42,43,44,45,46]. Finally allogeneic products can be obtained from a variety of sources, some of which might be expected to have more pluripotent and regenerative potential (e.g. umbilical cord blood, placental tissue), properties which might be used to treat conditions in which autologous products are not available (pediatric patients/congenital heart disease) or conditions which might require repair of multiple tissue types [47,48,49,50,51,52]. While these studies have been small, they have led the foundation to ongoing phase I-II studies which are scheduled to report in 2021 (NCT02549625, NCT02914171, NCT01883076, NCT02781922, NCT02398604). 

As concerns of rejection of allogeneic stem cells were allayed, especially in regard to mesenchymal stem cells or derivatives with mesenchymal stem cell properties [53,54,55], these cells were increasingly tested. While comparisons between allogeneic and autologous stem cells are few, when compared, allogeneic stem cells generally equaled or outperformed their autologous counterparts [24,56].

Finally, while governmental investment has led to the development of some therapeutics, industry is responsible for the vast majority of new treatments. In this regard, allogeneic cells, which represent a discrete regulatable, approvable, and potentially profitable product, have a distinct advantage over an autologous product in which elements of the purification process are largely the intellectual profit that would drive commercial development. 

### 4.2. Selected Cell Products

Bone marrow is easy to obtain and the mononuclear component can be simply and rapidly isolated; however, even its proponents recognize key limitations. Some of these limitations include the variable activity of bone marrow cells from older patients and those with vascular disease which lack the functional capacity of cells derived from healthier subjects [21,41,42,43,44,45,46], and the relative impurity of the cell product, with at least 98% of the cells representing mature potentially inflammatory cells with no stem cell potential. Bone marrow cells with pluripotential and capacity for long-term expansion, hallmarks of true stem cells, are even rarer, likely composing 1 of every 10,000 bone marrow mononuclear cells. Both pre-clinical [57] and clinical [58] efforts have suggested that purification may improve efficacy. In mouse models, purified CD34+ cells are more efficacious as a purified product when compared with the same number of cells injected as unselected bone marrow cells [57].

Cell selection based on cell surface markers, other flow cytometric characteristics, or expansion have each been utilized to purify bone marrow for a variety of concentrated stem cell products (Table 1) [23,33,36,59,60,61,62,63,64,65,66,67,68]. 

The use of technologies to isolate and characterize specific cell types offers the opportunity for commercial development based on intellectual property surrounding cell isolation processes. Some of these technologies also explored the utility of stem cells derived from other tissue sources, including cardiac [70] and adipose tissue [71,72,73,74]. Cardiac sources might be thought to be particularly capable of cardiac repair, although these cells have also been used to treat a variety of non-cardiac conditions and likely mediate most of their effects via mechanisms similar to other mesenchymal-like cells [75]. In addition to ease (and desirability in many cases) of harvest, adipose tissue may represent a source of cells that are uniquely resistant to aging and degradation in functional capacity in the elderly and those with cardiovascular disease. 

An untoward consequence was a rush to clinical testing of a panoply of cell types and the conduct of many concurrent underpowered trials that resulted in predictably inconclusive results. Unfortunately, the consequence is that a series of “neutral” findings has dampened enthusiasm for funding in this field, and several products with provocative findings [61,63,64,69] have had their clinical development curtailed due to business considerations at the sponsor level. 

### 4.3. Debate Over Efficacy: Rise of the Meta-Analysis

The initial focus on myocardial infarction, the interest by multiple investigators and institutions, especially in Europe, to offer stem cell therapy trials, and the development of multiple different cell types as outlined above led to a series of generally similar trials. While each was largely underpowered to detect modest clinical benefits, the similarities in trial designs led the field to attempts to synthesize the totality of clinical experience (Table 2) [26,27,28,29,30,69,76,77,78,79,80,81,82,83,84,85]. 

Indeed, the number of meta-analyses may seem to have outnumbered the numbers of clinical trials performed [86] and there is even a review of meta-analyses [87]. In aggregate, a majority of these efforts suggested a modest benefit on imaging parameters of left ventricular function and favorable point estimates on clinical endpoints. A brief inspection of clinical outcomes from the largest of the individual studies [23,88] suggests that an efficacy signal is present. The trends, even if not statistically significant, pointed towards improvements in hard cardiovascular endpoints and the outcomes certainly allay any safety concerns. Thus, it is likely this line of research has uncovered an approach with at least some potential for a favorable efficacy/safety profile. Nonetheless, demonstrating this requires adequately powered experiments with well-established protocols and statistical analysis plans [32]. Whether or not any of these therapies will go beyond what has been done to date remains to be seen. 

### 4.4. Shifts in Mechanistic Understanding

Early pioneers envisaged therapies that would repopulate the heart with stem cell derived cardiomyocytes, reversing the tissue lost after myocardial infarction and potentially chronic ischemia/heart failure. The earliest therapies explored the use of myoblasts for this purpose in patients with transmural scar [89,90,91,92]. While these cells did not reach fruition in terms of clinical development, myoblasts were shown to be capable of in situ generating new contractile tissue which was found in one study to persist for years [93,94,95]. As the focus shifted to adult bone marrow cells and their derivatives, it became clear that unlike myoblasts, bone marrow cells were not retained in significant quantities, rarely if ever differentiated into myocardial tissues or supporting structures, and did not directly lead to creation of new myocytes [8,9]. Animal models suggested that myocardial recovery post-myocardial infarction might nonetheless be enhanced by cell therapy, even in the absence of all of these capabilities [73,96,97], and the effects of a growing number of cell types were replicated by “cocktails” (exosomes, individual growth factors, supernatants, culture medium) derived from cells in culture [75,97,98,99,100,101,102]. This led to a change in the understanding of the mechanism by which stem cells might exert their effects, namely a transference from cells providing “hardware” to “software”—the concept of paracrine effects [97]. This paradigm shift has been a difficult pill for some critics to swallow, and indeed it may be difficult to understand how simple application of a cocktail of growth factors at one point in time may alter cardiac reparative capacity. Interestingly, careful investigation has demonstrated that even though bone marrow cells are not retained in significant numbers, persistence of a small number of these cells may be critical. For instance, when bone marrow cells are embedded with a suicide switch which is active several weeks post-cell delivery, their effectiveness was abrogated [103]. Interestingly, this seems to be particularly related to long-term engraftment of endothelial/angiogenic precursors, suggesting that neovascularization may underlie a significant portion of the benefit of these cell types [104].

This change in our understanding of the mechanisms of repair raises the possibility of affecting myocardial regeneration through cell-free products, a line under current investigation. 

### 4.5. New Targets

While the original paradigm of persistent engraftment leading to cardiac myocyte replenishment lent itself to the treatment of acute large transmural myocardial infarction, advances in the care of these patients, improvements in outcomes amongst even the sickest of acute myocardial infarction presenters after initial stabilization, and the difficulties of treating an acute condition have conspired to gradually shift the focus away from regenerative approaches for this indication. Two key patient populations seem to be best served by cell therapies.

#### 4.5.1. Chronic Heart Failure

Chronic heart failure is a disease marked by high levels of inflammatory and thrombotic markers. While small initial studies with bone marrow cells were conducted [36,58], the angiogenic, anti-inflammatory, and immunomodulatory properties of mesenchymal stem cells and their derivatives have received increasing interest. Notably some of the largest clinical trials to date have produced promising results. The IxCELL-DCM trial demonstrated that macrophage/mesenchymal stem cell product could impact total congestive heart failure hospitalizations, an approvable endpoint, with a favorable point estimate for effect on both overall mortality (RR 0.25) and death/left ventricular assist device/transplant (RR = 0.66) [63]. Inexplicably, clinical development has seemingly halted despite these positive results. The MSC-HF trial also met its primary, albeit imaging, endpoint [105]. An interesting, although open-label, non-randomized, experience from a single center in Slovenia reported a benefit on 5-year mortality when patients with non-ischemic cardiomyopathy were treated with endothelial/hematopoietic CD34^+^ stem cells [68]. This work has been expanded to other indications including right ventricular failure [106]. We now await full results from two large trials that are likely to shape our enthusiasm for this approach: the DREAM-HF trial enrolled 537 patients in a double-blind sham procedure controlled trial with a primary endpoint of a hierarchical clustering of clinical endpoints [107], and the SCIENCE trial, which is enrolling patients at 6 European heart failure centers [73]. Unpublished preliminary data from the DREAM-HF trial indicates that the primary endpoint was not met, although the sponsor notes favorable effects on mortality and “hard” cardiac endpoints such as myocardial infarction and stroke [60]. 

The possibility that cells derived from the target organs might be especially suited to enact tissue-specific repair has resulted in investigations focused on cardiac-derived or differentiated cells. The CHART-1 trial utilized bone marrow cells cultured and expanded with a “cardiopoietic cocktail” selected to enhance expression of cardiac transcription factors. In 315 patients randomized to treatment or sham procedure, the impact on the primary outcome was statistically neutral although the point estimate was positive, and a post-hoc analysis suggested that a select group of patients might benefit [108,109]. The ALLSTAR trial investigated the use of allogeneic cardiosphere-derived stem cells on infarct size and left ventricular remodeling and function. While the primary endpoint was neutral and the trial was terminated early, there were interesting and positive effects on indices of left ventricular size and brain natriuretic peptide levels [62]. These same cells were studied in a pilot study of 25 patients with Duchenne muscular dystrophy, and were associated with improvements in myocardial scar size (injury), but again attesting to stem cells paracrine and wide-ranging effects, surprisingly impacted peripheral muscle function, setting the stage for a larger efficacy trial [65]. Mesenchymal stem cells when administered to an extremely advanced heart failure population receiving left ventricular assist devices did not impact the ability to be weaned from support, but had dramatic and statistically powerful effects on gastrointestinal bleeding, reflective of their anti-inflammatory properties [110]. Finally, c-kit cells, which were at one time touted to substantively lead to myocardial regeneration [111,112], were tested clinically in the SCIPIO trial. An early publication suggested surprisingly large improvements in left ventricular function [113]. Although some papers relating to the SCIPIO trial have been retracted [114], it is important to note that this was prompted by concerns over pre-clinical data, while the clinical findings have never been questioned. C-kit cells were recently studied in the National Heart Lung and Blood Institute-sponsored CONCERT trial, the results of which were recently presented [70]. While there were some suggestions of benefit with combination c-kit/mesenchymal stem cell therapy, it remains unclear whether combination therapy is a viable option given the small sample size and disparate effects on the multitude of endpoints studied [115].

#### 4.5.2. Refractory Angina

In retrospect, it is perhaps surprising that more energy has not focused on patients with refractory angina. The refractory angina population comprises patients who pose a high burden to the health care system, are highly symptomatic, are limited in their ability to perform even daily activities, have poor quality of life [116,117,118,119], and lack any reasonably effective treatment options [120]. In addition, the fundamental deficit in these patients, notably microvascular function, is what many of the cells studied in the clinical arena might be expected to target most effectively. The largest program to date was a series of three double-blind intramyocardial injection placebo-controlled studies with many common design elements [61,64,121,122]. Each trial, a phase I feasibility study [121], a phase II dose-finding study [61], and a prematurely terminated phase III study [64,122], showed greater benefit in the CD34^+^ cell-treated patients on exercise time, angina frequency, and clinical endpoints. An analysis of the ACT-34 CMI phase II study demonstrated statistically significant improvements in exercise times in the low-dose treatment arms, and a statistically significant improvement in mortality to two years in CD34^+^ cell-treated patients. An analysis combining individual patient data from all three studies showed highly significant effects on all three key endpoints: exercise capacity, angina frequency, and mortality [69]. Unfortunately, funding for further studies has not been forthcoming, although a recent study in microvascular angina is also promising [123]. Notably, bone marrow cells are being used to treat patients with refractory angina on a limited, but approved and reimbursed basis, in the Netherlands based on data suggesting cost-effectiveness of this therapy [124,125,126,127]. 

## 5. Critical Appraisal

A “second phase” of research has led to expansion of cell types and disease states targeted for regenerative efforts in the cardiovascular arena. Major developments include the acknowledgement that the original objective to replace lost cardiomyocytes with tissue spawned directly from administered cells was an overly simplistic understanding of the regenerative process. Nonetheless, the prospect that functional recovery might still be realized without direct myocyte differentiation, “the paracrine hypothesis,” remains. Larger phase II studies suggest that selected cells might improve clinical outcomes in patients with refractory angina and systolic heart failure, but disappointingly, follow-up trials do not appear to be imminent. In our opinion, the plethora of relatively small underpowered or poorly planned clinical trials, the behavior of some in the preclinical community [128], and retractions of papers in this field [114,129] have resulted in skepticism in the broader cardiology research [130,131] and clinical communities regarding this line of investigation. Coupled with uncertainties from regulators regarding approval processes [132] and opportunities for reimbursement, there is also diminishing interest from the investment community. We now look to next generation approaches to re-stimulate the excitement and promise of this field. 

## 6. The Road Ahead 

While the first decades of clinical work have focused on cell types that were easier to obtain, isolate, and purify, and ultimately appear to exert their effects largely through paracrine mechanisms, we highlight three lines of investigation that may fulfill the promise of replacement of myocardium or its support structures (Figure 2). 

### 6.1. Enhancing Bone Marrow Derived Cells

Short lifespan after administration remains a major limitation of stem cell approaches tried to date. Genetic modification of stem cells (mesenchymal stem cells or cardiac (c-kit+) stem cells via enhanced expression of anti-apoptotic/pro-survival factors such as Akt and Pim1 has been shown to enhance cellular “engraftment” and persistence after administration in rodent models compared with the corresponding unmodified cells [97,134]. In larger animal models, Pim1 expressing CSCs decreased fibrosis/scar tissue formation threefold, increased functional mass twofold compared with unmodified CSCs, and produced significant improvements in regional and overall contractility and left ventricular performance [135].

One other approach to enhancing reparative capacity of mesenchymal and cardiac stem cells is based on the understanding that significant signaling occurs between cell types in any tissue, including the heart. Based on significant preclinical data that suggests synergistic effects between mesenchymal and cardiac stem cells, the CONCERT study was conducted to explore for synergy between mesenchymal and c-kit^=^ cardiac stem cells in patients with ischemic heart failure. Although small and not definitive, this study did suggest that combination therapy had more consistent and numerically greater effects on indices of left ventricular function and clinical outcomes compared with either cell type alone [115]. 

An alternative combinatorial approach utilizing stepwise co-culture of c-kit^+^ cardiac cells, mesenchymal stem cells, and endothelial progenitor cells promoting assembly of 3-dimensional “CardioClusters” of defined size and cell ratio has been shown to result in a scaffold-free product with increased survival characteristics [136]. In vitro, CardioClusters protected cardiomyocytes against injury compared with any of the cell lines tested individually. When injected into mouse myocardial infarction models, CardioClusters improved ejection fraction, left ventricular strain, and indices of remodeling compared with individual cell lines [136]. Clinical translation now awaits, however, determining efficacy of combination therapies in the clinic is difficult given that differences between various active therapies is expected to be smaller than comparisons with placebo. 

### 6.2. True Myocardial Regeneration

Autologous myoblasts were first studied in preclinical models as early as the 1980s, representing the first attempt to replace lost myocardial tissue with new contractile tissue that could supplement the mechanical capacity of the heart. While this was never successfully developed clinically, individual cases in which long-term engraftment and generation of new tissue resulting in improvement in left ventricular function, demonstrated even years after cell injection, remind us that myocardial regeneration remains a goal worth pursuing [95]. 

The original ambitions of cellular regenerative approaches to replace lost myocytes with active contractile tissue is now the focus of new lines of attack, driven by the ability of multipotent stem cells to differentiate into cardiac myocytes. Several groups are exploring the use of pluripotent stem cells differentiated into myocyte lineages to replace lost myocardial tissue. Results so far have pointed in disparate directions. 

#### 6.2.1. Embryonic Stem Cells

Embryonic stem cells were first successfully derived in 1998 [137] and their differentiation into cardiomyocytes reported soon thereafter [138,139]. Pre-clinical transplantation occurred rapidly. While the direct injection of pluripotent embryonic stem cells appeared to be limited by issues with unregulated differentiation [140], injection of cells that were differentiated and selected under controlled conditions led to stable graft formation in mouse, rat, and guinea pig models, leading finally to work in non-human primates [133,141,142]. Observations in these experiments highlight specific challenges in this arena, notably arrhythmias and the need for immunosuppression, findings anticipated by observations from the original myoblast studies [89]. Arrhythmias observed with embryonic stem cells appear particularly concerning given their duration and persistence. While there is some indication that these arrhythmias eventually abate [142,143] (as was seen with myoblasts [89]), the tolerability of such arrhythmias in humans with significant heart failure and when or whether the propensity for arrhythmia would abate in humans and its responsiveness to anti-arrhythmic therapies/procedures remains to be determined. 

In addition, this allogeneic approach has required significant immunosuppression in animal models. Proponents point to the fact that immunosuppression is fundamental to the viability of the field of organ transplant [143], and the success of long-term engraftment and generation of contractile tissue fulfills the initial goals of this field [141]. Whether or not myocyte replacement via such approaches can mimic the benefit or replace/defer heart transplantation remains to be determined.

#### 6.2.2. Induced Pluripotent Stem Cells

An alternative pluripotent source for cardiomyocyte differentiation originates from the transformational technology described less than 15 years ago by Yamanaka et al, in which differentiated cells (usually fibroblasts) are reprogrammed to an undifferentiated embryonic state via culture with four growth factors (Oct3/4, Sox2, c-Myc, and Klf4) [144]. The use of allogeneic induced pluripotent stem cell-derived cardiomyocytes has also garnered attention and may serve as an alternative to embryonic stem cell-derived derived cardiomyocytes [145]; however, as with embryonic stem cell-derived cardiomyocytes, induction of ventricular arrhythmias remains a concern. 

Unlike their embryonic counterparts, induced pluripotent stem cell-derived cardiomyocytes could serve as an autologous or major histocompatibility complex-(partially) matched source. Autologous induced pluripotent stem cells, in which cardiomyocytes might be derived from a patient’s own induced pluripotent stem cells, might obviate the need for immunosuppression and serves as a theoretically attractive means to regenerating a patient’s own cardiomyocytes. However, this approach is limited by the cost and time required to reprogram, isolate, and expand the billions of cardiomyocytes that constitute a therapeutic dose, a process that is measured in months. Studies in non-human primates have shown that major histocompatibility complex-matched allogeneic induced pluripotent stem cell-derived cardiomyocytes might not only require less immunosuppression, but may demonstrate enhanced efficacy compared with non-major histocompatibility complex-matched allogeneic cardiomyocytes [146]. The prospect of “universal donor” induced pluripotent stem cells, in which gene editing technologies have been used to alter or abolish major histocompatibility complex expression, offers an alternative approach to mitigating risk of donor rejection or cell loss [147,148]. The feasibility of these approaches will likely be dictated by whether an analogy of “Moore’s law” in the technology field applies to the costs and methods of induced pluripotent stem cell technology development, and to what degree enhanced efficiencies in this regard might make this approach competitive with transplantation. One strategy which may offer simplicity and greater efficiency is direct fibroblast reprogramming, which has garnered significant preclinical attention [149,150,151,152]. In addition to more efficient generation of cardiomyocytes for direct administration, these approaches are exploring generation of multipotent but cardiac committed progenitors, as well as in vivo reprogramming of fibroblast and other cardiac cells to enact cellular regeneration without direct stem cell administration [153]. 

### 6.3. Mixed Products and Bioengineered Constructs

While regeneration of human cardiac tissue via direct injection of stem cells or stem cell-derived cardiomyocytes into the human heart represents the ultimate goal of this field, combining a bioengineering approach may offer a speedier more direct pathway to clinical application [154]. Several groups have fabricated patches from induced pluripotent stem cell- or embryonic stem cell-derived cardiomyocytes. The small proof-of-concept ESCORT study (NCT02057900) demonstrated the feasibility of constructing a fibrin patch embedded with human embryonic stem cell-derived cardiac progenitors and “cell-delivery” during coronary artery bypass surgery [155]. No arrhythmias were observed. Under short-term immunosuppression with cyclosporine and mycophenolate, low levels of anti-donor antibodies were detected without clinical manifestation. Improvements in functional and imaging endpoints were reported; however, given these patients all underwent revascularization, the efficacy of the patch remains speculative. Long-term imaging of the patch was not reported. Pre-clinical research from Menasche’s lab suggests that many if not all the benefits of this approach might be due to paracrine effects [156].

What is becoming clearer is that the multitude of hurdles to the development of functional and clinically impactful myocardial patches are gradually being overcome. Advances are being made in adequate cell isolation; expansion and growth of patches large enough for large animal testing and eventual human use; and adequate maturation of cardiomyocytes into structures that mirror those in native myocardium both structurally and functionally, including excitability and contractility, action potential conduction, calcium handling, and proper cardiomyocyte size and function [154,157,158,159]. As with many of the findings in the field of cell therapy, much of the improvement observed in several of these studies is felt to be due to cell secretomes, and not a direct action of the cells themselves [156]. An interesting approach to a cell-free patch approach has recently been reported, suggesting that even cellular bioengineering approaches may find clinical application through a cell-free route [160].

### 6.4. Microvascular Disease: A Final Common Pathway?

As cardiovascular regenerative medicine enters its third decade of clinical research, one observation is increasingly evident: with the exception of attempts at direct embryonic stem cell- or induced pluripotent stem cell-derived cardiomyocyte injection, the initial premise that we could achieve cardiomyocyte replacement and repletion is false. The initial endorsement of a “paracrine” hypothesis may have been unsurprising when applied to unselected bone marrow populations with limited transdifferentiating potential, but appears to equally apply to selected cells, “pluripotent” mesenchymal cells, cardiac stem cells, and now even embryonic stem cell- and induced pluripotent stem cell-derived patches, all approaches initially felt to represent avenues to myocardial regeneration. 

Multiple lines of research suggest effects on the microvasculature might represent a “final common pathway” for many of the regenerative approaches to date. Regardless of the absolute degree to which host cells generated new myocytes in transplanted hearts, there was a consistent finding that endothelial and smooth muscle cells of host origin were more commonly identified, indicating higher levels of stem cell-mediated vascular regeneration [16,18]. Virtually all of the cell types utilized in clinical studies, including “cardiomyogenic” cells, have been shown to have angiogenic and neovascularization capabilities in pre-clinical models [57,97,161,162,163,164]. Notably, even in patients with “non-ischemic” cardiomyopathy, cell therapy which improved outcomes was associated with improvements in measures of perfusion [165].

Microvascular dysfunction is increasingly recognized as a common pathway in a variety of manifestations of cardiovascular disease, including chronic angina and acute coronary syndromes in patients both with and without epicardial coronary disease. Similarly, it is critical to the development of left ventricular and biventricular dysfunction in ischemic, diabetic, hypertensive, and likely other forms of “non-ischemic” cardiomyopathies. The mechanisms that lead to microvascular dysfunction, including fibrotic, inflammatory, apoptotic, and endothelial autonomic dysfunction, are all targets of stem cell therapy, largely through production of growth factors, cytokines, chemokines, and other factors that directly counteract each of these mechanisms [166].

Excluding work with direct injection of embryonic stem cells and induced pluripotent stem cells, we proffer the opinion that endothelial repair may underlie a substantial fraction of any beneficial effects of cell therapies studied to date. We tender this as an opinion to stimulate discussion and further experimentation, recognizing that controversy about mechanisms of action surround therapies that have been in clinical use for decades. We note the following:(1)The cells under consideration do not result in long-term engraftment or directly generate new myocardial tissue [8,9,11].(2)Replacement of vascular cells occurs with much greater frequency in transplanted organs than differentiated myocytes [16,17,18].(3)Pre-clinical studies demonstrate that the predominant effects of stem cells in ischemic models includes enhancement in vascular and capillary growth and decreases in fibrosis [57,167].(4)Cell therapy has been demonstrated to provide improvement in microvascular parameters in both pre-clinical models and patients [167,168,169]. The recent demonstration of efficacy in patients with microvascular angina in which endothelial dysfunction is the primary culprit is particularly thought provoking [123].(5)Cells committed to angiogenic lineages may have enhanced effects on improvements in cardiac function [103,104].(6)Highly angiogenic cells seem to be associated with greater potency. The data with CD34+ cells in refractory angina [69], congestive heart failure [68], and microvascular angina [123] are particularly compelling, given the angiogenic nature of these cells.(7)While the effect of cell therapy on composite endpoints has been somewhat variable, the point estimate for mortality consistently favors cell therapy in congestive heart failure [63,68,77,105] and refractory angina [69]. Impairments in endothelial function directly predict mortality in a variety of cardiovascular conditions. Effects on microvascular function offer a potential mechanistic link between preclinically demonstrated benefits of current cell therapies [57] and clinical observations.

## 7. Summary

The field of cardiovascular cell therapy is in its third decade and is a more mature area of investigation with more measured expectations and a greater understanding of goals for the field. The replacement of lost cardiomyocytes with stem cell-derived tissue has proven difficult to achieve in the short term; nonetheless, it remains an aspirational achievement worth pursuing. We believe next generational approaches to this goal will arise from three avenues of research. First, cells derived from undifferentiated sources, including induced pluripotent and embryonic stem cells, will be explored as sources of new cardiomyocytes in quantities sufficient to replace those lost in both acute and chronic human disease. Second, combining cell therapy with bioengineered matrices/devices may generate biological constructs aimed at enhancing myocardial function and performance. Finally, the lessons learned in the previous decades of research will act as the foundation for approaches whose goals are now better understood to focus on myocardial salvage via paracrine mechanisms, a process we believe is largely mediated via vascular repair. 

## Figures and Tables

**Figure 1 cells-10-00600-f001:**
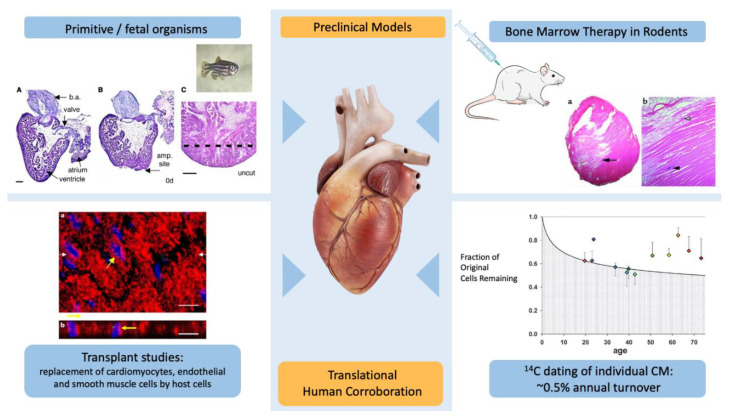
Evidence Supporting Possibility of Stem Cell Medicated Cardiac Repair. Top left: Primitive organisms such as zebrafish are able to regenerate significant (~25%) of resected myocardium with histologically identical myocardium. Adapted with permission [2]. Top right: early reports of use of bone marrow cells suggested significant trans-differentiation and generation of new myocardium, reports which were frequently difficult to reproduce. Adapted with permission [4]. Bottom right: Evidence of human myocardial regeneration based on ^14^C myocyte content suggests turnover of ~0.5% per year which decreases with age. Adapted from with permission [13]. Bottom left: A series of studies suggested replacement of transplanted endothelial, vascular, and myocyte tissue by host cells, although the rate of myocardial replacement was in most cases low (~0.04%). Adapted from with permission [14].

**Figure 2 cells-10-00600-f002:**
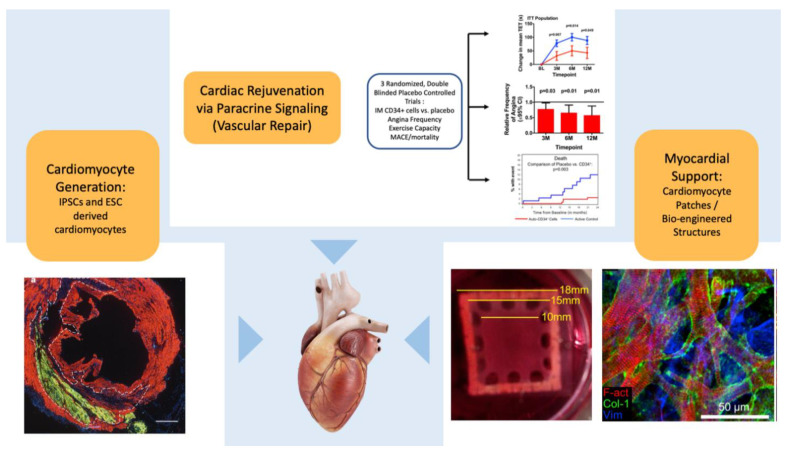
Future Approaches for Cardiac Regeneration/Rejuvenation: Top: Use of selected cells to enhance myocardial repair and rejuvenation to enhance vascular repair, reduce microvascular dysfunction, exert anti-inflammatory effects, and improve cellular function via paracrine effects. As one example, CD34^+^ cells may have significant clinical effects in patients with refractory angina even in the absence of myocardial generation [69]. Lower right: Confluence of bioengineering/cell therapeutic approaches to construct patches which may promote regeneration through direct mechanical and paracrine mechanisms. The left figure shows a 12-day patch of rat cardiomyocytes, the right panel shows staining for f-actin (red), collagen (green), and vimentin (purple). Adapted with permission from Jackman et al, Biomaterials, 159: 48-58 (2018). Lower left: Pluripotent stem cell sources such as cardiomyocytes derived from embryonic stem cells can permanently engraft and proliferate in the heart (green staining demonstrates green fluorescent protein expressing cardiomyocytes in non-human primate infarct model (Macaque monkey) at day 14). Adapted with permission [133].

**Table 1 cells-10-00600-t001:** Key Clinical Trials of Bone Marrow and Selected Cells for Cardiovascular Disease.

Selection/Characteristics	Key Clinical Trials	Disease State	Key Endpoints	Outcomes
**Unselected BM Cells**				
Ficoll gradient	REPAIR-AMI [23]	Post-AMI	Δ EF	Small increase in EF (2–5%), variably reproducible
Sepax closed automated processing system	FOCUS [36]	HFrEF	Improvement in LV dimensions, perfusion, or peak MVO2	Powered for large treatment effect, study neutral
Sepax closed automated processing system	TIME and LATE-TIME [66,67]	Post-AMI	Δ EF	U.S. multisite trials, no effect on EF post-AMI
Ficoll gradient	BAMI [33]	Post-AMI	Mortality	Prematurely terminated, Expected: 12%; Observed: 3.5%
**Cardiosphere-Derived Stem Cells**				
Cardiac progenitor derived from cardiac explant cultures	ALLSTAR [62]	Post-AMI	Δ scar size	Prematurely terminated, positive effects on BNP and remodeling
	HOPE [65]	DMD	Δ myocardial scar size	Improvements in muscle strength in DMD
**C-kit+**				
SC of cardiac neural crest origin	SCIPIO	Ischemic CM	Δ EF post-CABG	Partial analysis published
	CONCERT	HFrEF	Variety of exploratory endpoints	Possible synergistic effects with MSCs in CHF
**CD34+**				
Angioblast marker: hematopoietic and endothelial stem cells	ACT-34 and RENEW [61,64,69]	Refractory angina	Improvements in angina frequency, exercise time in refractory angina	Improvements, RENEW prematurely terminated by sponsor, mortality improved
	Vrtovec et al. [68]	Non-ischemic cardiomyopathy	Mortality, BNP	Improvements (open-label study)
**CD133+**				
Primitive (hematopoietic) stem cell marker	Several small trials [59]	Post-AMI	Δ EF	Signals of (small) improvements
**Allogeneic mesenchymal stem cell**				
Multipotent stem cells isolated based on adherence and growth on plastic	DREAM-HF [60]	HFrEF	Combined clinical endpoint	Unpublished, enrollment curtailed from 1800 to <600
**Ixmyelocel-T (CD90+, CD14+auto+M2 cells)**				
Pluripotent mesenchymal stem cells with anti-inflammatory macrophages	ixCELL-DCM [63]	HFrEF	CHF admissions, mortality	Lower risk of CHF events

AMI indicates acute myocardial infarction; BM, bone marrow; BNP, brain natriuretic peptide; CABG, coronary artery bypass graft; CHF, congestive heart failure; CM, cardiomyopathy; DMD, Duchenne muscular dystrophy; EF, ejection fraction; HFrEF, heart failure with reduced ejection fraction; LV, left ventricular; MSCs, mesenchymal stem cells; MVO2, myocardial volume oxygen; SC, stem cells.

**Table 2 cells-10-00600-t002:** Summary of meta-analyses of Bone Marrow Cell Therapy for Ischemic Heart Disease.

	Population	Cells	Studies	No. of Patients	EF HR (95% CI)	Mortality	CHF MACE or CHF Hospitalizations	Ischemic MACE or MI	Other
**Myocardial Infarction**									
Hristov et al. (2006) [79]	AMI	BMMC	5	482	4.21 (0.21, 8.22) *p* = 0.04				
Abdel-Latif et al. (2007) [26]	IHD	BMMC, BMMesC, CPCs	18	807	3.64 (1.56, 5.73) *p* < 0.001	No Δ			↓LVEDD, ↓ LVESD, ↓ IS
Lipinski et al. (2007) [82]	AMI	BMMC	10	698	3.0 (1.9, 4.1) *p* < 0.001	OR 0.52 (0.16, 1.63) *p* = 0.26	OR 0.32 (0.09, 1.21) *p* = 0.09	OR 0.22 (0.05, 0.90) *p* = 0.04	Trend ↓LVEDD, ↓ LVESD, ↓ IS
Martin-Rendon et al. (2008) [83]	AMI	BMMC	13	811	2.99 (1.26, 4.72) *p* = 0.0007	RR 0.62 (0.22, 1.76) *p* = 0.37	RR 0.61 (0.12, 2.96) *p* = 0.54		↓ LVESD, ↓ IS
Jeevanantham et al. (2012) [27]	IHD	BMMC, BM-MSCs,	50	2625	3.96 (2.90, 5.02) *p* < 0.00001	OR 0.39 (0.27, 0.55) *p* < 0.00001	OR 0.52 (0.27, 1.00) *p* = 0.06	OR 0.25 (0.11, 0.57) *p* = 0.001	36 RCT, 14 cohort studies, ↓LVEDD, ↓ LVESD
Zimmet et al. (2012) [85]	AMI	BMMCs	23	1317	2.70 (1.48, 3.92) *p* < 0.001	OR 0.64 (0.22, 1.72) *p* = 0.46	OR 0.62 (0.16, 2.20) *p* = 0.59	RR = 0.66 (0.16, 2.45) *p* = 0.67	↓LVEDD, ↓ LVESD
Delewi et al. (2013) [28]	AMI	IC BMMC	24	1624	2.23 (1.00, 3.47) *p* = 0.004	RR 0.60 (0.34, 1.08) *p* = 0.09	RR 0.59 (0.35, 0.98) *p* = 0.04	RR 0.44 (0.24, 0.79) *p* = 0.007	No Δ in LVEDD or LVESD
de Jong et al. (2014) [29]	AMI	BMMC, BM-MSCs			2.10 (0.68, 3.52) *p* = 0.004	OR 0.68 (0.36, 1.31) *p* = 0.25	OR 0.14 (0.03, 0.52) *p* = 0.003	OR 0.5 (0.24, 1.06) *p* = 0.07	↓ LVESD, ↓ IS
Xu et al. (2014) [84]	IHD	BMMC, CPCs	19	886	3.54 (1.92, 5.17) *p* < 0.001	RR 0.49 (0.28, 0.84) *p* = 0.01		RR 0.29 (0.06, 1.53) *p* = 0.14	No Δ LVEDD, ↓ LVESD
Fisher at al. (2016) [76]	AMI	BMMC, BM-MSC, CD34+ or CD133+ cells	41	2739	0.27 (-1.13, 1.67) *p* = 0.70	HR 0.92 (0.62, 1.36) *p* = 0.67	HR 0.36 (0.21, 0.61) *p* = 0.002	HR 0.63 (0.40, 1.01) *p* = 0.05	
Gyongyosi et al. (2015) [78]	AMI	IC BMMC	12	767:485	1.15 (-0.38, 2.69) *p* = 0.14		HR 0.81 (0.57, 1.16) *p* = 0.25	HR 0.52 (0.28, 1.08) *p* = 0.08	No Δ in LVEDD or LVESV, patient level analysis
**Ischemic Cardiomyopathy**									
Kandala et al. (2013) [80]	HFrEF (ICM)	BMMC	10	519	4.48 (2.43, 6.53) *p* < 0.0001				↓LVEDD, ↓ LVESD
Fisher et al. (2015) [77]	HFrEF (ICM)	BMMC, CPCs,	31	1521	2.06 (1.1, 3.01) *p* < 0.0001	RR 0.48 (0.34, 0.69) *p* < 0.0001	RR 0.39 (0.22, 0.70) *p* = 0.002		Multiple different study types, cells, delivery mechanisms, and additional procedures (PCI/CABG) at time of cell delivery
**Refractory Angina**									
Fisher et al. (2013) [30]	CIHD	BMMC and CD34+ Cells	9	659		RR 0.33 (0.17, 0.65) *p* = 0.001			↓CCS angina class, ↓ AF
Li et al. (2013) [81]	CIHD	BMMC and CD34+ Cells	5	381		OR 0.33 (0.08, 1.39) *p* = 0.13		OR 0.37 (0.14, 0.95) *p* = 0.04	↑ETT (61.3 s [18.1, 104.4] *p* = 0.005; ↓ AF
Henry et al. (2018) [69]	CIHD	CD34+ Cells	3	304		K-M rate 2.5% (CD34+) vs. 12.1% (placebo) *p* = 0.0025		30.0% (CD34+) vs. 38.9% (Placebo) *p* = 0.14	↑ETT (49.5 s [9.3, 89.7] *p* = 0.016; RR AF 0.66 (*p* = 0.012)

AF indicates angina frequency; AMI, acute myocardial infarction; BMMC, bone marrow mononuclear cells; BMMesC, bone marrow derived stem cells; BM-MSC, bone marrow mesenchymal cells; CABG, coronary artery bypass graft; CCS, Canadian Cardiovascular Society; CHF, congestive heart failure; CIHD, chronic ischemic heart disease; CPC, cardiac progenitor cells; EF, ejection fraction; ETT, exercise tolerance test; HFrEF, heart failure with reduced ejection fraction; IC, intracoronary; IHD, ischemic heart disease; ICM, ischemic cardiomyopathy; IS, infarct size; K-M, Kaplan-Meier; LVEDD, left ventricular end-diastolic diameter; LVESD, left ventricular end-systolic diameter; MACE, major adverse cardiovascular events; MI, myocardial infarction; PCI, percutaneous coronary intervention; RCT, randomized controlled trial; RR, Relative Risk.

## Data Availability

Not applicable.

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
