# Peer review of "Stem Cells in Cardiovascular Diseases: 30,000-Foot View"

_cells, 2021, doi:10.3390/cells10030600_

Round 1
Reviewer 1 Report
This is a nicely organized and reasonably impartial assessment of cell therapy for myocardial disease primarily focused upon regenerative therapy approaches in the cardiac context. For the most part the authors have done a respectable job of arbitrating the controversies and disappointments in the historical retrospective of clinical trials. I do have a few comments to provide more balanced and comprehensive coverage of the main points:
1) The “Future directions” section mentions “next generation approaches” but does not mention two important avenues for improving outcomes: 1) combinatorial cell therapy, and 2) use of modified cells for enhanced cell therapy.
Two relevant references and an appropriate review article are as follows:
Monsanto MM, Wang BJ, Ehrenberg ZR, Echeagaray O, White KS, Alvarez R, Fisher K, Sengphanith S, Muliono A, Gude NA, Sussman MA Enhancing myocardial repair with CardioClusters. Nat Commun 2020 08 07 11(1):3955. Epub 2020 Aug 07
Broughton KM, Sussman MA Enhancement Strategies for Cardiac Regenerative Cell Therapy: Focus on Adult Stem Cells. Circ Res 2018 07 06 123(2):177-187.
Kulandavelu S, Karantalis V, Fritsch J, Hatzistergos KE, Loescher VY, McCall F, Wang B, Bagno L, Golpanian S, Wolf A, Grenet J, Williams A, Kupin A, Rosenfeld A, Mohsin S, Sussman MA, Morales A, Balkan W, Hare JM Pim1 Kinase Overexpression Enhances ckit Cardiac Stem Cell Cardiac Repair Following Myocardial Infarction in Swine. J Am Coll Cardiol 2016 Dec 06 68(22):2454-2464.
2) The term “remuscularization” to describe the primate models receiving ES or iPS cells as coined in the original studies is misleading. Please revise to “persistent engraftment” as a more accurate and specific representation.
Author Response
Reviewer #1: This is a nicely organized and reasonably impartial assessment of cell therapy for myocardial disease primarily focused upon regenerative therapy approaches in the cardiac context. For the most part the authors have done a respectable job of arbitrating the controversies and disappointments in the historical retrospective of clinical trials. I do have a few comments to provide more balanced and comprehensive coverage of the main points:
1) The “Future directions” section mentions “next generation approaches” but does not mention two important avenues for improving outcomes: 1) combinatorial cell therapy, and 2) use of modified cells for enhanced cell therapy.
Two relevant references and an appropriate review article are as follows:
Monsanto MM, Wang BJ, Ehrenberg ZR, Echeagaray O, White KS, Alvarez R, Fisher K, Sengphanith S, Muliono A, Gude NA, Sussman MA Enhancing myocardial repair with CardioClusters. Nat Commun 2020 08 07 11(1):3955. Epub 2020 Aug 07
Broughton KM, Sussman MA Enhancement Strategies for Cardiac Regenerative Cell Therapy: Focus on Adult Stem Cells. Circ Res 2018 07 06 123(2):177-187.
Kulandavelu S, Karantalis V, Fritsch J, Hatzistergos KE, Loescher VY, McCall F, Wang B, Bagno L, Golpanian S, Wolf A, Grenet J, Williams A, Kupin A, Rosenfeld A, Mohsin S, Sussman MA, Morales A, Balkan W, Hare JM Pim1 Kinase Overexpression Enhances ckit Cardiac Stem Cell Cardiac Repair Following Myocardial Infarction in Swine. J Am Coll Cardiol 2016 Dec 06 68(22):2454-2464.
Thank you. We have added the following to the “Road Ahead” section of our manuscript, highlighting the possibility of using combination therapies to enhance effectiveness.
Short lifespan after administration remains a major limitation of stem cell approaches tried to date. Genetic modification of stem cells (mesenchymal stem cells or cardiac (c-kit+) stem cells via enhanced expression of anti-apoptotic/pro-survival factors such as Akt and Pim1 has been shown to enhance cellular “engraftment” and persistence after administration in rodent models compared with the corresponding unmodified cells [97,133]. In larger animal models, Pim1 expressing CSCs decreased fibrosis/scar tissue formation threefold, increased functional mass twofold compared with unmodified CSCs, and produced significant improvements in regional and overall contractility and left ventricular performance [134].
One other approach to enhancing reparative capacity of mesenchymal and cardiac stem cells is based on the understanding that significant signaling occurs between cell types in any tissue, including the heart. Based on significant preclinical data that suggests synergistic effects between mesenchymal and cardiac stem cells, the CONCERT study was conducted to explore for synergy between mesenchymal and c-kit= cardiac stem cells in patients with ischemic heart failure. Although small and not definitive, this study did suggest that combination therapy had more consistent and numerically greater effects on indices of left ventricular function and clinical outcomes compared with either cell type alone [115].
An alternative combinatorial approach utilizing stepwise co-culture of c-kit+ cardiac cells, mesenchymal stem cells, and endothelial progenitor cells promoting assembly of 3-dimensional “CardioClusters” of defined size and cell ratio has been shown to result in a scaffold-free product with increased survival characteristics [135]. In vitro, CardioClusters protected cardiomyocytes against injury compared with any of the cell lines tested individually. When injected into mouse myocardial infarction models, CardioClusters improved ejection fraction, left ventricular strain, and indices of remodeling compared with individual cell lines [135]. Clinical translation now awaits, however, determining efficacy of combination therapies in the clinic is difficult given that differences between various active therapies is expected to be smaller than comparisons with placebo.
2) The term “remuscularization” to describe the primate models receiving ES or iPS cells as coined in the original studies is misleading. Please revise to “persistent engraftment” as a more accurate and specific representation.
We have changed the sentence to read “While the original paradigm of persistent engraftment leading to cardiac myocyte replenishment”
see attachment for full responses.

Reviewer 2 Report
In this appealing review the authors recapitulate the stem cell-based approaches so far exploited in some cardiovascular disease settings. They clearly explain possible reasons that led from enthusiastic preclinical findings to often inconclusive or disappointing clinical results with an encouraging look to future developments. Despite the manuscript is objective and well written, some minor issues need to be addressed.
-Page 7: table 2 is difficult to read, please consider reformatting.
- Please, to avoid confusion, differentiate the table footnotes using a smaller font size than the one used for the main text.
-Page 12 lines 347-351 the authors state: “Finally, c-kit cells….were tested clinically in the SCIPIO trial. An early publication suggested surprisingly large improvements in left ventricular function [108]. Although this paper has been retracted [109], it is important to note that this was 350 based on concerns over pre-clinical data and not on the basis of concerns over the clinical findings”, I suggest not to use retracted papers in support of the clinical benefit of such approach. Along the same line, given that the paper relative to the SCIPIO trial has been retracted, I also suggest to remove it from table 1.
-The second and third avenues of research hypothesized in the summary section seem to contradict each other, please revise.
-Ref 35 and ref 109 refer to the same paper, please amend.
Author Response
Reviewer #2:
In this appealing review the authors recapitulate the stem cell-based approaches so far exploited in some cardiovascular disease settings. They clearly explain possible reasons that led from enthusiastic preclinical findings to often inconclusive or disappointing clinical results with an encouraging look to future developments. Despite the manuscript is objective and well written, some minor issues need to be addressed.
We appreciate the reviewers comments.
-Page 7: table 2 is difficult to read, please consider reformatting.
- Please, to avoid confusion, differentiate the table footnotes using a smaller font size than the one used for the main text.
The table footnotes are in a significantly smaller font size, so we are unclear as to what the reviewer is requesting. We have made some changes to the headings in the table to make it easier to understand.
-Page 12 lines 347-351 the authors state: “Finally, c-kit cells….were tested clinically in the SCIPIO trial. An early publication suggested surprisingly large improvements in left ventricular function [108]. Although this paper has been retracted [109], it is important to note that this was 350 based on concerns over pre-clinical data and not on the basis of concerns over the clinical findings”, I suggest not to use retracted papers in support of the clinical benefit of such approach. Along the same line, given that the paper relative to the SCIPIO trial has been retracted, I also suggest to remove it from table 1.
Thank you for this observation. We also struggled with whether or not to use these references. We have updated the references to include the 2-year follow-up which has not been retracted, We again emphasize that the clinical results have never been questioned.
-The second and third avenues of research hypothesized in the summary section seem to contradict each other, please revise.
Thank you. We have revised the summary section to more closely align with the way these points were raised and discussed in the rest of the manuscript.
-Ref 35 and ref 109 refer to the same paper, please amend.
Thank you, we have fixed this oversight.
Please see attachment for full response to all reviewers

Reviewer 3 Report
This review by Drs. Povsic and Gersh provides a comprehensive overview of stem cell approaches in preclinical and clinical therapies for cardiovascular disease. It is well written and easy to follow. Current progress is nicely summarized in the tables. I have some suggestions for the authors to consider.
This review was focused mainly on bone marrow-derived progenitor cells. Other cell types were not discussed or only briefly mentioned in the table. Preclinically, multiple cell types such as induced pluripotent stem cells, umbilical cord cells, mesenchymal stem cells, and bone marrow-derived progenitor cells have shown therapeutic potential in animal models of cardiovascular diseases. In clinical trials for CHDs, cardiosphere-derived cells are being tested for treatment of single ventricle physiology, bone marrow-derived mesenchymal stem cells, or umbilical cord blood cells for hypoplastic left heart syndrome and bone marrow-derived mononuclear cell for pediatric dilated cardiomyopathy. It worth elaborating a bit further in the text.
The figure panels are too small and not legible. Since the details of some published figure panels are not necessary for this review, it might be better to use graphic abstracts (or schematic cartoons) instead.
Page 3, line 90. It would be helpful to define what hardware and software are (e.g., in brackets).
Induced pluripotent stem cells were discussed in section 6.1.2. It worth mention that direct reprogramming of fibroblasts into functional cardiomyocytes is a major preclinical approach that is under intensive studies.
Author Response
Reviewer #3:
This review by Drs. Povsic and Gersh provides a comprehensive overview of stem cell approaches in preclinical and clinical therapies for cardiovascular disease. It is well written and easy to follow. Current progress is nicely summarized in the tables. I have some suggestions for the authors to consider.
This review was focused mainly on bone marrow-derived progenitor cells. Other cell types were not discussed or only briefly mentioned in the table. Preclinically, multiple cell types such as induced pluripotent stem cells, umbilical cord cells, mesenchymal stem cells, and bone marrow-derived progenitor cells have shown therapeutic potential in animal models of cardiovascular diseases. In clinical trials for CHDs, cardiosphere-derived cells are being tested for treatment of single ventricle physiology, bone marrow-derived mesenchymal stem cells, or umbilical cord blood cells for hypoplastic left heart syndrome and bone marrow-derived mononuclear cell for pediatric dilated cardiomyopathy. It worth elaborating a bit further in the text.
We appreciate the reviewer’s input and concerns. We agree that many topics were not completely covered in this review. We attempted to give a “30,000 foot overview” of the field’s development and future, focusing on the areas which have consumed the greatest energy and resources and generated the most controversy and attention.
We have elaborated about the use of various sources of stem cells with theoretically enhanced pluripotent potential for a variety of uses including hypoplastic left heart syndrome, pediatric cardiomyopathy, and congenital heart disease as follows (page 8):
Finally allogeneic products can be obtained from a variety of sources, some of which might be expected to have more pluripotent and regenerative potential (e.g. umbilical cord blood, placental tissue), properties which might be used to treat conditions in which autologous products are not available (pediatric patients / congenital heart disease) or conditions which might require repair of multiple tissue types [47-52]. While these studies have been small, they have led the foundation to ongoing phase I-II studies which are scheduled to report in 2021 (NCT02549625, NCT02914171, NCT01883076, NCT02781922, NCT02398604).
The figure panels are too small and not legible. Since the details of some published figure panels are not necessary for this review, it might be better to use graphic abstracts (or schematic cartoons) instead.
Page 3, line 90. It would be helpful to define what hardware and software are (e.g., in brackets).
We have condensed two sentences into the following: “Whether or not the signaling pathways (software) and cellular components (hardware) that allow this type of regenerative capacity can be adequately characterized, reactivated and to what extent they are even present in humans remains to be determined.”
Induced pluripotent stem cells were discussed in section 6.1.2. It worth mention that direct reprogramming of fibroblasts into functional cardiomyocytes is a major preclinical approach that is under intensive studies.
Thank you. We have added the following on page 25 of the manuscript:
One strategy which may offer simplicity and greater efficiency is direct fibroblast reprogramming, which has garnered significant preclinical attention [149-152]. In addition to more efficient generation of cardiomyocytes for direct administration, these approaches are exploring generation of multipotent but cardiac committed progenitors, as well as in vivo reprogramming of fibroblast and other cardiac cells to enact cellular regeneration without direct stem cell administration [153].
Please see attachment for full response to all reviewers.
